# Intent to Vaccinate SARS-CoV-2 Infected Children in US Households: A Survey

**DOI:** 10.3390/vaccines9091049

**Published:** 2021-09-21

**Authors:** Mehgan Teherani, Samridhi Banskota, Andres Camacho-Gonzalez, Alison G. C. Smith, Evan J. Anderson, Carol M. Kao, Charles Crepy D’Orleans, Andi L. Shane, Austin Lu, Preeti Jaggi

**Affiliations:** 1Department of Pediatrics, Emory University School of Medicine, Children’s Healthcare of Atlanta, Atlanta, GA 30322, USA; acamac2@emory.edu (A.C.-G.); evanderson@emory.edu (E.J.A.); ashane@emory.edu (A.L.S.); 2Emory University School of Medicine, Atlanta, GA 30322, USA; samridhi.banskota@emory.edu (S.B.); alison.smith@emory.edu (A.G.C.S.); austin.lu@emory.edu (A.L.); 3Department of Pediatrics, Washington University School of Medicine, St. Louis, MO 63110, USA; kaoc@wustl.edu

**Keywords:** vaccine hesitancy, pediatrics, children, COVID-19, SARS-CoV-2, mRNA, intent-to-vaccinate

## Abstract

A paucity of data exists evaluating a guardian’s intent to vaccinate their child against COVID-19 in the United States. We administered 102 first (April–November 2020) and 45 second (December–January 2020–2021) surveys to guardians of children (<18 years) who had a laboratory-confirmed diagnosis of COVID-19 and assessed their intent to give a COVID-19 vaccine to their child, when one becomes available. The first and second surveys of the same cohort of guardians were conducted before and following the press releases detailing the adult Pfizer-BioNTech and Moderna Phase 3 results. Both surveys included an intent-to-vaccinate question using the subjective language of “if a safe and effective vaccine” became available, and a second question was added to second surveys using the objective language of “would prevent 19 of 20 people from getting disease”. When using subjective language, 24 of 45 (53%) guardians endorsed vaccine administration for their children in the first survey, which decreased to 21 (46%) in the second survey. When adding objective language, acceptance of vaccination increased to 31 (69%, *p* = 0.03). Common reasons for declining vaccination were concerns about adverse effects and/or vaccine safety. Providing additional facts on vaccine efficacy increased vaccine acceptance. Evidence-based strategies are needed to increase pediatric COVID-19 vaccine uptake.

## 1. Introduction

As of 9 September 2021, approximately 5.3 million children have been diagnosed with SARS-CoV-2 infection with pediatric cases rising exponentially since the early summer months [1]. While studies at the beginning of the pandemic suggested that children contributed less to the transmission of SARS-CoV-2 than adults, additional studies describe children as important contributors to COVID-19 transmission, especially as in-person school has returned [2,3,4]. Therefore, in order to successfully achieve community protection from COVID-19 disease, vaccination in children will be crucial [5,6,7]. Rapid development, manufacturing, and running of human clinical trials for COVID-19 vaccines have resulted in Emergency Use Authorization (EUA) by the United States (US) Food and Drug Administration (FDA) of the Pfizer-BioNTech vaccine in young adolescents 12 years of age and above, with clinical trials underway for those 6 months and older [ClinicalTrials.gov Identifier: NCT04816643].

Prior to the SARS-CoV-2 pandemic, the World Health Organization (WHO) listed vaccine hesitancy as a top 10 threat to global health [8]. Before EUA of the three currently approved COVID-19 vaccines in adults, intent-to-vaccinate among adults has ranged from 58–69% [9,10,11,12]. An international cross-sectional survey found that caregiver intent to vaccinate their children against SARS-CoV-2 was approximately 65% [13]. In addition, parents in households with children were less likely to accept a COVID-19 vaccine for themselves compared to households without children [14,15]. To understand guardian intent-to-vaccinate their COVID-19 recovered child before and after the press releases describing the phase 3 efficacy of the Moderna and Pfizer-BioNTech Phase 3 COVID-19 vaccines in adults [16,17], we sought to identify factors contributing to vaccine hesitancy in order to optimize strategies to improve COVID-19 vaccine uptake by guardians for their children.

## 2. Materials and Methods

We identified children (<18 years old) at random with a laboratory-confirmed SARS-CoV-2 infection by PCR of a nasopharyngeal (NP) swab between 16 March and 14 September 2020. Asymptomatic children with a positive SARS-CoV-2 test were excluded.

Up to three attempts were made to contact families and surveys were conducted in Spanish or English at least 14 days (1 incubation period) after the date of the child’s first positive COVID-19 test. A guardian of each child was first interviewed from March 16-September 14 by telephone using a 19-question survey to evaluate their child’s symptom duration, the timing of household contacts with symptoms consistent with COVID-19, or had a laboratory-confirmed COVID-19 test, and patient and household members’ symptom severity [18]. The first survey also asked three questions regarding the acceptability of a potential SARS-CoV-2 vaccine (Figure 1). Responses were recorded in a secure, online database (REDCap) by the interviewer [18]. A follow-up survey was then conducted among the same guardians from 9 December 2020–6 January 2021, following the press releases reporting >90% vaccine efficacy for both mRNA COVID-19 vaccines, Pfizer-BioNTech on 9 November 2020 and Moderna on 16 November 2020. The second survey included the same 3 intent-to-vaccinate questions in the initial survey plus an additional re-phrased question that included additional efficacy data (Figure 1). The second survey also assessed the guardian’s source of vaccine information (news versus social media versus other) and if the information seen was pro- or anti-vaccine (Appendix A).

Descriptive statistics were reported as total numbers, percentages, and medians with interquartile ranges (IQR) as appropriate. Descriptive statistics and statistical testing were performed in R (R Core Team, 2017). We performed Wilcoxon rank-sum tests for comparison of continuous variables, and chi-square and Fisher’s exact tests were used to compare categorical variables.

The Georgia Immunization Registry (GRITS), which mandates electronic documentation of all non-investigational vaccines administered in Georgia within 30 days of the date of administration, was used to document the immunization status of the children of the respondents.

This study was reviewed by the local Children’s Hospital Institutional Review Board (#00000594) and verbal consent was obtained by the patient’s legal guardian.

## 3. Results

The guardians of 1143 children aged 2–15 years old, with laboratory-confirmed acute SARS-CoV-2 were eligible to participate. We initially called families in order of positivity, however, as cases rose exponentially, we began the random selection of 201 guardians found in our RedCap database. Three contact attempts were made by telephone between 27 April–25 November 2020 and we completed 102 first surveys (51%). Of the 102 legal guardians from who we obtained the first surveys, 45 of 102 (44%) were successfully reached for second surveys from 9 December 2020–6 January 2021. The median follow-up time from positive test to completion of the initial survey was 37 (IQR 26–51) days. The median age of the SARS-CoV-2 positive children surveyed was 8 (IQR 4–14) years and 59% were male. Significantly more children’s guardians identified as Hispanic ethnicity and Black race in those surveyed compared to all patients with COVID-19 disease, *p* = 0.01 and *p* = 0.03, respectively (Table 1). There were significantly more hospitalized patients interviewed compared to those who were not interviewed, *p* = 0.03 (Table 1).

Of the initial 102 guardians surveyed, 46 (45%) endorsed COVID-19 vaccination of their child, while 44 (43%) would not, and 12 (12%) might (Figure 1). We did not find significant differences in patient demographics or a child’s COVID-19 disease severity in guardians who would or would not vaccinate their child (Table 2). Comparing the 45 respondents to both the first and second surveys, 24 (53%) guardians in the first and 21 (46%) in the second survey would support COVID-19 vaccination of their child (Figure 1). Sixteen of 45 (36%) respondents to the first survey would refuse COVID-19 vaccination of their child, while 14 (31%) of the second survey respondents would refuse (Appendix A). Five (11%) first respondents might support COVID-19 vaccination of their child, while 10 (22%) might on the second survey. Reasons for “no” or “maybe” responses are detailed in Table 3.

In the second survey, legal guardians were also asked a more specific (“yes” or “no” only) question, if they would allow administration of a COVID-19 vaccine to their child, which prevents 19 of 20 people from getting COVID-19 disease. Of 44 asked this question, 31 (70%) said “yes” while 14 (30%) said “no”.

Thirty- two of the 45 follow-up survey respondents (71%) had reported reading pro- or anti-vaccine news reports. Ten were aware of pro-vaccine reports, five were aware of anti-vaccine reports, and 17 reported they were aware of both types of reports. Significantly more families would allow a vaccine to be given to their child if they had heard pro-vaccine reports compared to anti-vaccine or both types of reports (80% vs. 0% vs. 47%, *p* = 0.031). There was no significant difference in willingness to vaccinate their child among guardians who reported social media as their news source.

In follow-up surveys, 4 of 45 (9%) of guardians changed their opinion from “no to yes” in allowing the vaccine to be given to their child. Nine (20%) changed their opinion from “no to yes” to indicate agreement that vaccinating their child would prevent others from getting infected, while 1 (2%) respondent changed their opinion from “yes to no” (Appendix A).

Forty-three of 44 children with immunization records in GRITS were completely vaccinated for age. Of those children who received seasonal influenza vaccination within the last 2 seasons (2019–2020), 21 of 29 (72%) said “yes or maybe” to administering the COVID-19 vaccine to their child, whereas 9 of 14 (64%) who did not receive the influenza vaccine said “yes or maybe” (*p* = 0.58, NS).

## 4. Discussion

Among guardians of children with a preceding medical encounter for acute COVID-19, approximately 50% were vaccine-hesitant despite widely publicized efficacy and safety results from the Pfizer-BioNTech and Moderna phase 3 clinical trials. We would expect that given guardians who completed surveys had children with more severe illness, families would be more willing to accept a vaccine, however, our findings are congruent with the CDC’s recent report of 48% of 16–17 year-olds and 39% of 12–15 year-olds who are fully vaccinated in the US as of 8 September 2021 [19,20]. In a recent study conducted in South Korea, guardians had a higher reported intent to vaccinate their child at 64.2%, mirroring their confidence in the effectiveness of COVID-19 vaccines of 64.8% [21]. In our study, adding a positive efficacy fact to the question in the second survey, appeared to influence guardians’ intent to vaccinate their child. We also found a lack of association between patients who had received influenza vaccination and willingness to receive a COVID-19 vaccine.

Pediatricians play an important role in counseling families on COVID-19 vaccines for children, as studies have shown that parents are more likely to trust their own doctor or healthcare provider than the media, pharmaceutical companies, and government organizations [22,23]. In addition, a recent national survey found that 50% of parents have not discussed COVID-19 vaccination with their child’s provider, but that discussion with their provider is important for their decision to vaccinate their child [24]. A systematic review and meta-analysis found no single type of communication improved uptake [23]. However, our survey found that adding positive vaccine information increased vaccine acceptance (45% vs. 69%) and may be a helpful strategy for Pediatricians to use during patient visits.

Common reasons for reported COVID-19 vaccine refusal included concerns about the vaccine side effects, which parallels the findings of prior studies of adults intent-to- vaccinate themselves [14,22,25]. While we found that hearing pro-vaccine news greatly improved public perception of vaccination and hearing anti-vaccine news reduced intent-to-vaccinate, we were unable to determine causality, as guardians with a higher propensity to be vaccine-hesitant may already seek out anti-vaccine information. In a study evaluating “YouTube”, investigators found that 27% of COVID-19 related videos had no scientific evidence but had over 60 million views [26]. In another study performed by Blankenship, et al., anti-vaccine tweets on Twitter were 4.13-fold more likely to be “re-tweeted” than neutral tweets [27]. We did not find a significant difference between different news platforms and willingness to vaccinate. We suspect that although social media platforms lack regulation and editorial oversight, in general, various news sources are not well-trusted by the public [28,29]. While social media companies have taken initial measures to curb false vaccine-related information, such as hiding false reports and disrupting bots, there is still a lack of accountability and transparency and allows for rapid dissemination of misinformation [30]. The CDC has provided recommendations for critical stakeholders to aid in providing clear information to the public about vaccines [31].

Limitations to our study include not having a control cohort of children not infected with COVID-19 enrolled concomitantly, which would have further strengthened the generalizability of our conclusions. However, in a previous study, there were no differences in rates of vaccine hesitancy of parents of children who had COVID-19 disease versus those that did not [32]. In addition, we did not find significant differences in patient demographics and intent to vaccinate their child, which may have been due to our small sample size, the higher proportion of persons of the Black race that make up the composition of the Atlanta metropolitan area, and our pediatric patient population. In addition, we had relatively low response rates to the first and second surveys, however, our findings are similar to current vaccination rates in US children aged 12–17 [20]. Similar to our overall COVID-19 cohort during our study period, a significant proportion of our study participants were of Hispanic or Latino ethnicity, which may limit our generalizability to other geographic areas in the US. The timing of the second survey may have also impacted responses as it was conducted at the onset of mass COVID-19 vaccination of the adult public, but before EUA approval in children, when additional safety and efficacy data became available.

## 5. Conclusions

Intention to refuse the COVID-19 vaccine is prevalent in about half of caregivers of children with COVID-19. Specific language and use of positive facts to educate families may improve guardian intent-to-vaccinate but will likely need a diverse range of targeted evidence-based interventions [33]. Monitoring reasons for vaccine refusal and approval in households with children will aid in finding objective methods to improve guardian vaccine attitudes and achieve community protection against COVID-19.

## Figures and Tables

**Figure 1 vaccines-09-01049-f001:**
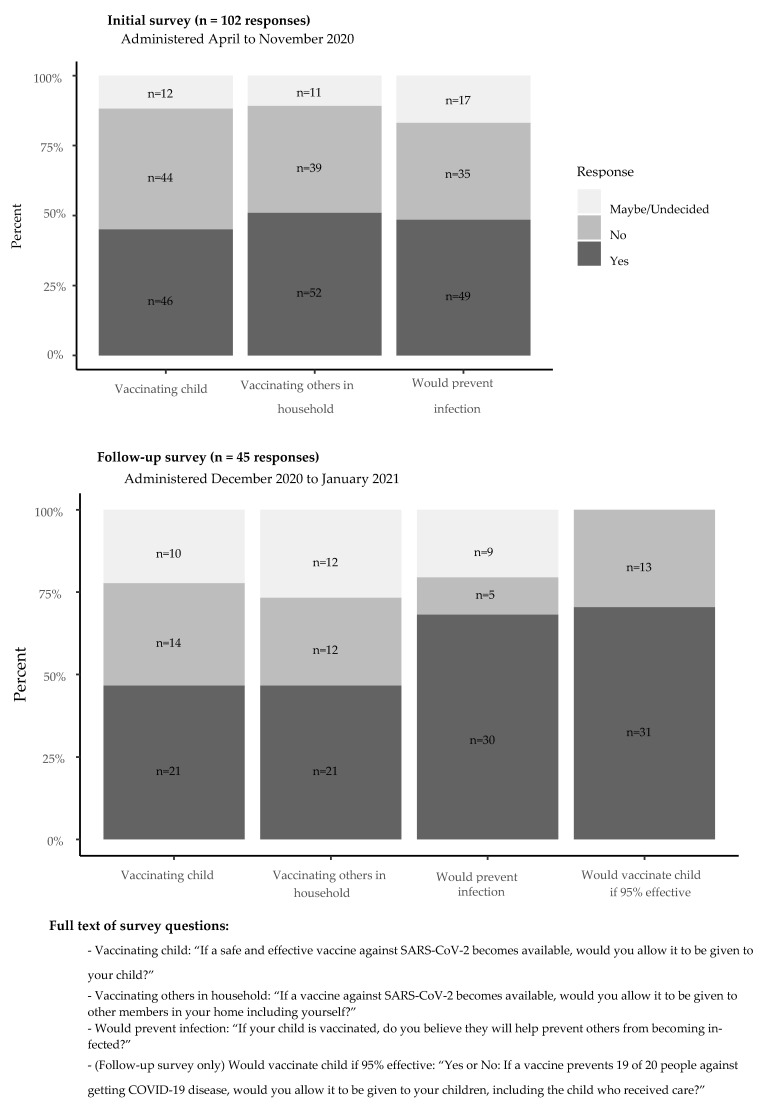
Guardian vaccine acceptability on first and second surveys.

**Table 1 vaccines-09-01049-t001:** Patient demographics: all pediatric COVID19 patients vs. surveyed guardians (first) vs. surveyed guardians (second).

Characteristic	All Pediatric COVID-19 Patients, N = 1143	All Patients Whose Guardians Were Surveyed, N = 102 ^1^	*p*-Value ^2^	Guardian Completed First Survey Only, N = 57 ^1^	Guardian Completed First and Second Surveys, N = 45 ^1^	*p*-Value ^2^
Age of patient, years (median, IQR)	9 (2, 15)	8 (4, 14)	0.8	8 (2, 14)	9 (5, 14)	0.8
*Unknown*	1	*10*		*7*	3	
Sex of patient (male)	591 (52%)	60 (59%)	0.2	28 (49%)	32 (71%)	0.04
Ethnicity: Hispanic or Latino	368 (35%)	44 (46%)	0.01	24 (45%)	20 (48%)	0.2
*Unknown*	*81*	*7*		*4*	*3*	
Race: White	441 (39%)	49 (48%)	0.08	26 (46%)	23 (51%)	0.7
Race: Black	406 (36%)	25 (25%)	0.03	15 (26%)	10 (22%)	0.8
Obesity		8 (12%)		6 (16%)	2 (7.4%)	0.5
*Unknown*		*38*		*20*	*18*	
Other comorbidities ^3^		4 (6.2%)		2 (5.4%)	2 (7.4%)	>0.9
*Unknown*		*38*		*20*	* 18 *	
Another household member had symptoms consistent with COVID-19		79 (77%)		44 (77%)	35 (78%)	>0.9
Location patient was seen			0.03			0.8
*Inpatient*	217 (19%)	29 (28%)		15 (26%)	14 (31%)	
*Emergency Department (ED)*	566 (50%)	37 (36%)		23 (40%)	14 (31%)	
*Urgent care*	346 (30%)	34 (33%)		18 (32%)	16 (36%)	
*Other*	14 (1.2%)	2 (2.0%)		1 (1.8%)	1 (2.2%)	
Time from positive test to date of the primary survey, days (median, IQR)		37 (26, 51)		38 (28, 49)	29 (24, 68)	0.5
*Unknown*		20		* 13 *	* 7 *	

^1^ Statistics presented: Median (IQR); n/N (%). ^2^ Statistical tests performed: Wilcoxon rank-sum test; chi-square test of independence; Fisher’s exact test. ^3^ Comorbidities included: Type 1 diabetes, type 2 diabetes, seizure disorders, congenital heart disease, sickle cell disease, chronic lung disease, congenital malformations, and other comorbidities non-specified.

**Table 2 vaccines-09-01049-t002:** Patient demographics by whether parent or guardian was vaccine-hesitant * at time of the primary survey.

Intent to Vaccinate Their Child	Yes, N = 46 ^1^	Maybe or No, N = 56 ^1^	*p*-Value ^2^
Age of patient, years (median, IQR)	8 (2, 12)	9 (4, 14)	0.4
*Age unknown*	7	3	
Age, categorical			0.2
*<12 years*	28 (72%)	30 (57%)	
*≥12 years*	11 (28%)	23 (43%)	
*Unknown*	7	3	
Sex of patient (male)	27 (59%)	33 (59%)	>0.9
Ethnicity: Hispanic/Latino	23 (52%)	21 (41%)	0.2
*Ethnicity unknown*	2	5	
Race: White	22 (48%)	27 (48%)	>0.9
Race: Black or African American	8 (17%)	17 (30%)	0.2
Child with obesity	1 (3.8%)	7 (18%)	0.13
*Unknown*	20	18	
Location where patient was seen			0.5
ED or Urgent Care	35 (76%)	38 (68%)	
Hospitalized (inpatient)	11 (24%)	18 (32%)	

* Vaccine hesitancy defined as a “maybe” or “no” response to the question, “If a safe and effective vaccine against SARS-CoV-2 becomes available, would you allow it to be given to your child?”. ^1^ Statistics presented: Median (IQR); n (%). ^2^ Statistical tests performed: Wilcoxon rank-sum test; chi-square test of independence; Fisher’s exact test.

**Table 3 vaccines-09-01049-t003:** Reasons for “no” or “maybe” in giving COVID-19 vaccine to their child on secondary *.

Number of Reports (*n* = 24)	Reasons
12	Safety and side effect concerns
9	Lack of information
4	Can give you COVID-19 illness
3	Not 100% effective
3	Heard people have died from the vaccine
2	Does not get other vaccines, including influenza vaccine
2	Development of vaccine was too rapid
2	Made from human fetuses
1	Child has antibodies after COVID-19 infection
1	Vaccine will modify your genes
1	Heard a baby had brain damage after receiving the vaccine
1	Religious reason

* Some guardians stated more than one reason for vaccine refusal.

## Data Availability

The data presented in this study are available on request from the corresponding author. The data are not publicly available due to maintaining Health Information Privacy in accordance with Children’s Healthcare of Atlanta Instiutitonal Review Board.

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
