# Peer review of "Intent to Vaccinate SARS-CoV-2 Infected Children in US Households: A Survey"

_vaccines, 2021, doi:10.3390/vaccines9091049_

Round 1
Reviewer 1 Report
I appreciate that the authors undertook the survey. As the authors point out at the end of the manuscript, the sample size is very small and, apparently, geographically concentrated in one State in the United States, so it is not clear how externally valid the results can be. Also, there is a lot of noise about whether those who have been infected with COVID-19 require vaccination or whether vaccination is recommended. Could some of this confusion be contributing to the initial and follow-up answers?
Author Response
Point 1: “As the authors point out at the end of the manuscript, the sample size is very small and, apparently, geographically concentrated in one State in the United States, so it is not clear how externally valid the results can be.”
Response 1: We agree with the reviewer and believe that a larger multi-institutional study is needed. However, the current overall rate of fully vaccinated pediatric patients in the US approximate our findings with 43% of 16-17 year-olds and 33% of 12-15 year-olds according to the American Academy of Pediatrics and the Centers for Disease Control and Prevention as of August 19, 2021.
Point 2: “Also, there is a lot of noise about whether those who have been infected with COVID-19 require vaccination or whether vaccination is recommended. Could some of this confusion be contributing to the initial and follow-up answers?”
Response 2: We do acknowledge that previous infection could be a confounding factor in our cohort, however, when we asked respondents about reasons they said “no” or “maybe” to vaccination for their child, only 1 guardian responded that the child had antibodies after COVID-19 infection (described in Table 3). A majority of our patients reported safety and side effect concerns, and lack of information as the primary reasons for not intending to vaccinate their child. In addition, a previous study by He, K., et. al. found that a child having COVID-19 infection did not affect desirability of a guardian to give a COVID-19 vaccination to their child. We have added this to our Discussion, paragraph 4.
Reviewer 2 Report
This study reports the acceptance and hesitancy of parents towards COVID-19 vaccination for their children. Decisions for vaccination in children is challenging with the substantially rapid developmental process and limited safety data. It is important to assess the parents' insights into the vaccine for implementation. The reviewer has some comments as following.
- The authors performed the analysis on parents of children who were previously diagnosed with COVID-19. Considering that the families had experienced the disease itself, this factor may influence the acceptance rate (hesitancy rate) for the vaccine. It would be desirable to compare the acceptance rate with other previous reports among children in the discussion and speculate the impact of previous infection.
- Also, was there difference in intent to vaccinate among parents of children admitted to the hospital compared with children treated in the emergency department or urgent care?
- Was there a difference in acceptance rate according to the child's age and other demographic findings? Please include data and the authors opinion on this aspect.
- This survey was performed relatively early, before vaccination in the general adult population was initiated and even more before vaccination in children was done. It shows early responses towards the COVID-19 vaccines. 1) Please compare responses between recent reports after vaccination has been authorized in children 2) It is interesting that there were parents who were willing to vaccinate their children even before efficacy and safety reports were announced in the first survey. What do the authors think could have influenced the parents decisions? Also have the authors looked into the parents "reasons for saying yes to vaccination"? If so, what were the thoughts of the parents?
Author Response
Point 1: The authors performed the analysis on parents of children who were previously diagnosed with COVID-19. Considering that the families had experienced the disease itself, this factor may influence the acceptance rate (hesitancy rate) for the vaccine. It would be desirable to compare the acceptance rate with other previous reports among children in the discussion and speculate the impact of previous infection.
Response 1: Thank you for this input. In looking at a previous study by He, K., et. al., we found that a child having COVID-19 infection did not affect desirability of a guardian to give COVID-19 vaccination to their child. We have added this reference to the Discussion, paragraph 4. In addition, when we asked respondents about reasons they said “no” or “maybe” to vaccination for their child, only 1 guardian responded that the child had antibodies after COVID-19 infection (described in Table 3). A majority of our patients reported safety and side effect concerns, and lack of information as the primary reasons for not intending to vaccinate their child.
Point 2: Also, was there difference in intent to vaccinate among parents of children admitted to the hospital compared with children treated in the emergency department or urgent care?
Response 2: We did not find a difference in intention to vaccinate among parents of children admitted to the hospital compared to children being treated in the emergency department or urgent care. We have added Table 2 as additional analysis of this potential confounding variable.
Point 3: Was there a difference in acceptance rate according to the child's age and other demographic findings? Please include data and the authors opinion on this aspect.
Response 3: Although this has been seen in adult studies, we did not find a difference in age or other demographic variables. We have included this data in Table 2. We do not believe there was a difference in age as EUA had not yet been approved vaccine for pediatric patients at the time of this survey. We did not find a significant difference in Hispanic ethnicity or Black race, possibly due to the make-up of the Atlanta metropolitan area, which is 50% Black race, in addition to our small sample size.
Point 4: This survey was performed relatively early, before vaccination in the general adult population was initiated and even more before vaccination in children was done. It shows early responses towards the COVID-19 vaccines. 1) Please compare responses between recent reports after vaccination has been authorized in children 2) It is interesting that there were parents who were willing to vaccinate their children even before efficacy and safety reports were announced in the first survey. What do the authors think could have influenced the parents decisions? Also have the authors looked into the parents "reasons for saying yes to vaccination"? If so, what were the thoughts of the parents?
Response 4: 1) Unfortunately, there is a paucity of data evaluating intent-to-vaccinate children following EUA of Pfizer-BioNTech COVID-19 vaccine for those 12-17 years of age in the US. A study in South Korea found guardian intent-to-vaccinate their child percentage of 64.2%, however, given the variations in government trust, type of vaccines available, and patient demographic and socioeconomic differences, we do not feel that this is generalizable to the US population. We have added this study to the Discussion, paragraph 1. 2) We believe that parents were willing to accept a vaccine for their child as our question stated “if a safe and effective vaccine” became available and did not imply that they would need to administer the vaccine without this known safety and efficacy data. We did not evaluate reasons for parents saying yes to vaccination, but agree with the reviewer that this would be interesting to evaluate and analyze. We have added this to the conclusions for future directions.